# Looking at the fringes of MedTech innovation: a mapping review of horizon scanning and foresight methods

Sonia Garcia Gonzalez-Moral ,[1] Fiona R Beyer,[1] Anne O Oyewole,[2] Catherine Richmond,[1] Luke Wainwright ,[2] Dawn Craig[1]

¹NIHR Innovation Observatory at Population Health Sciences Institute, Newcastle University, Newcastle upon Tyne, UK
²Population Health Sciences Institute, Newcastle University, Newcastle upon Tyne, UK

**Correspondence to**
Sonia Garcia Gonzalez-Moral; sonia.garcia-gonzalez-moral@newcastle.ac.uk

## ABSTRACT

**Objectives** Horizon scanning (HS) is a method used to examine signs of change and may be used in foresight practice. HS methods used for the identification of innovative medicinal products cannot be applied in medical technologies (MedTech) due to differences in development and regulatory processes. The aim of this study is to identify HS and other methodologies used for MedTech foresight in support to healthcare decision-making.

**Method** A mapping review was performed. We searched bibliographical databases including MEDLINE, Embase, Scopus, Web of Science, IEEE Xplore and Compendex Engineering Village and grey literature sources such as Google, CORE database and the International HTA database. Our searches identified 8888 records. After de-duplication, and manual and automated title, abstracts and full-text screening, 49 papers met the inclusion criteria and were data extracted.

**Results** Twenty-five single different methods were identified, often used in combination; of these, only three were novel (appearing only once in the literature). Text mining or artificial intelligence solutions appear as early as 2012, often practised in patent and social media sources. The time horizon used in scanning was not often justified. Some studies regarded experts both as a source and as a method. Literature searching remains one of the most used methods for innovation identification. HS methods were vaguely reported, but often involved consulting with experts and stakeholders.

**Conclusion** Heterogeneous methodologies, sources and time horizons are used for HS and foresight of MedTech innovation with little or no justification provided for their use. This review revealed an array of known methods being used in combination to overcome the limitations posed by single methods. The review also revealed inconsistency in methods reporting, with a lack of any consensus regarding best practice. Greater transparency in methods reporting and consistency in methods use would contribute to increased output quality to support informed timely decision-making.

## INTRODUCTION

Horizon scanning (HS), understood as a systematic examination of information sources to detect early signs of important developments,[1] is a reliable method to

## STRENGTHS AND LIMITATIONS OF THIS STUDY

⇒ This review updates the methodological knowledge on horizon scanning practice for health and care decision-making of medical technologies (MedTech) innovations.
⇒ This is the first time a systematic literature search is expanded beyond traditional horizon scanning methods to identify other foresight methods applied in MedTech innovation detection and prediction.
⇒ Quality appraisal of the included studies was deemed out of scope and was not performed.

inform legal, regulatory or procurement decisions on new and emerging health technologies.[1 2] HS is a well-established method to support early awareness systems for new and emerging medicines;[3] wider uses of HS include some early work applied to models of care.[4] This paper focuses on the use of HS for emerging innovative medical technologies (MedTech) detection and prioritisation. The strategic importance of HS is recognised and endorsed at the highest level; for example, in the UK, the 2016 independent Accelerated Access Review recommended that the National Health Service (NHS) should develop an enhanced HS process to identify products that have the most potential to deliver improved outcomes or efficiencies, acknowledging a more comprehensive and transparent HS system is necessary, especially for non-pharmaceutical products. This task was mandated to the NHS Accelerated Access Collaborative (AAC) by the Secretary of State in 2018. The 2019 'NHS Long Term Plan' sets out to accelerate innovation and patient access to innovative therapies.[5] More recently, the 2021 Government's 'Life Sciences Vision' policy statement aims to stimulate industry growth and success for the benefit of patients and the NHS.[6] Both high-level national policies advocate for the use of HS to achieve their goals, a sentiment that is echoed in the Government's 'Saving and Improving Lives:

The Future of UK Clinical Research Delivery' policy paper (2021) and associated Implementation Plan (2022) which reinforces the AAC's commitment 'to improve identification of the most needed treatments and technologies and rapidly bring these into clinical use' through HS.

Foresight and HS methods share similar concepts and purpose but differ as follows. The objectives of foresight activities can vary across projects but generally aim to open discussion and debate in anticipation of strategy formulation with the aim of equipping organisations navigating an uncertain future. In HS, however, the aim is the identification of emerging issues or signals and their targets and connections within to elicit a plan of action.[7] The Innovation Observatory (IO) is the HS and research intelligence organisation funded by the National Institute for Health and Care Research and hosted by Newcastle University in the UK. Based on our experience in the practice of HS in the context of healthcare innovations, we observe that well-conducted HS is a complex and time-consuming process that requires a range of skills and a depth of knowledge to facilitate the identification, interpretation and filtration of relevant signals, in addition to highly developed technological tools to support the processing and management of data and knowledge and facilitate its dissemination in a timely manner. Other analytical foresight methods such as patent, bibliometric and trend analyses as well as non-analytical methods such as backcasting, roadmaps or Delphi studies are available that could complement or help solve some of these challenges when applied to the identification and prediction of MedTech innovation, but to these authors' knowledge, there is no evidence to support their use in any specific scenario.

MedTech (including devices, diagnostics and digital health applications and systems) have the potential to transform care pathways, improve health and contribute to cost efficiencies in the healthcare system.[8] HS methods used to signal new medicinal products cannot be directly translated and applied to MedTech due to the challenges posed by the intrinsic characteristics of these technologies' development and regulatory pathways. These challenges are wide ranging and multilayered; however, for the purposes of this paper, we will concentrate on those that affect the identification of these technologies ahead of their placement on the market. First, in order to discern where the true value for the healthcare system lies, a common definition and interpretation of 'innovation' is needed.[9] Innovation in MedTech may encompass a completely new device class or newer iterations of the same product.[10] Second, different regulators of MedTech follow different criteria and requisites for their assessment. These differences have an impact on how and when information is shared with regulators, and whether it can be made available to the wider public. In turn, this has an impact on the ability of HS searches to retrieve and identify signals of commercial readiness. For example, while the US Food and Drug Administration maintains medical devices databases that can be freely accessed,[11]

the European Union regulator has not made this information available to the wider public yet.[12] Third, traditional published literature sources might not fully contribute to identify emerging or early innovative MedTech. While this is also the case in HS practice for innovative medicinal products, for MedTech, grey literature, mainly originated by manufacturers, acts as the main source in up to 50% of the time.[13]

Previous reviews into HS methods in healthcare have been undertaken;[1 14] however, none of them have looked at how these methods perform when applied to the MedTech innovation pipeline. This review aims to build from what is already known by broadening the scope to identify foresight methods applied to MedTech innovation identification or prediction, including but not limited to HS; and to explore the drivers that lead to the application of these methodologies in the context of MedTech to gain an insight into the rationale for choosing a particular method over others. This work is the first step for devising a methodological framework to guide foresight practice in MedTech innovation decision-making.

## METHODS
In line with the core principles for conducting mapping reviews outlined by the Campbell Collaboration,[15] this review aimed to identify evidence of use of HS and foresight methods for MedTech innovation detection. A protocol was agreed among the review team which guided the inclusion and exclusion of studies as well as the systematic literature searching in databases and grey literature sources. This review did not undertake quality appraisal of the included studies as this was considered out of scope. Preferred Reporting Items for Systematic Reviews and Meta-Analyses extension for reporting systematic literature searches was followed and reported in online supplemental table A.1.[16]

### Patient and public involvement
Members of the public were not involved in this review.

### Inclusion criteria
Studies were eligible for inclusion if: (1) any of the following methods were mentioned in the text: environmental scanning, HS, foresight, forecasting, roadmapping, patent, bibliometric or trend analyses, backcasting or Delphi studies; (2) a medical health technology such as a medical device, diagnostic, digital intervention or all were included; (3) published in English.

### Search strategy
The search strategy consisted of a combination of subject headings and text words in title, abstracts and keyword fields. The strategy was peer reviewed by an independent information specialist using the Peer Review of Search Strategies (PRESS) review statement.[17] The final MEDLINE strategy was translated and run separately in: Embase, Scopus, Web of Science, IEEE Xplore and

Compendex Engineering Village. Grey literature sources (Google and CORE[18]) and International Horizon Scanning organisations (via International HTA Database) were searched. No time or language limits were used. Detailed search strategies are included in online supplemental appendix B. Reference lists of included papers were browsed to identify potential papers missed. Search results were downloaded to EndNote V.20 (Clarivate Analytics) and de-duplicated.

### Data collection
Results were de-duplicated and screened in two stages using Rayyan, a semiautomated screening tool for systematic reviews.[19] At stage one, the main reviewer screened the title and abstracts; with each vote for inclusion or exclusion, Rayyan produced a five-star ranking for all the references uploaded. A random sample of 100 low-ranked (three stars or less) references were screened to assess the consensus of the semiautomated ranking with the reviewer's manual screening decisions.

At second stage, two reviewers manually double-screened the full texts and selected those that met the eligibility criteria. Disagreements were reconciled by discussion between the two reviewers.

### Data extraction
A custom data extraction form was designed in Excel to record the relevant characteristics of the included papers. The extracted data captured: (1) author name and publication year, (2) title, (3) objective of the article, (4) study type, (5) methods used, (6) sources used, (7) technology type, (8) time horizon, (9) type of output produced and (10) automated methods used.

### Data synthesis
Extracted data were further categorised by objectives described and information on methods used to allow for analysis as some discrepancies in the reporting were present. Data were synthesised visually, tabulated and in narrative formats.

## RESULTS
Database and grey literature searches identified 8888 records. After de-duplication, 7782 remained. The first screening pass included a total of 228 references (more details on this stage on online supplemental figure C.1). Two reviewers manually screened the title and abstract of these and excluded a further 111 references. At second stage, 117 full-text papers were manually double-screened, concluding with 49 papers published between 1999 and 2020 that met the inclusion criteria. Preferred Reporting Items for Systematic Reviews and Meta-Analyses flow chart is included in online supplemental figure C.1.

### Analysis of objectives
Ten different categories of aims and objectives were identified (figure 1).

### Methods and sources
The included studies reported 25 single different methods. Twenty-nine studies used a single method[20–45] and 20 used multiple methods in combination.[46–65] The methods that overlap between the group of papers that reported using one method and the group that used more than one in combination were: expert panels, patent searches/analyses, literature searches, stakeholders, HS, roadmapping,

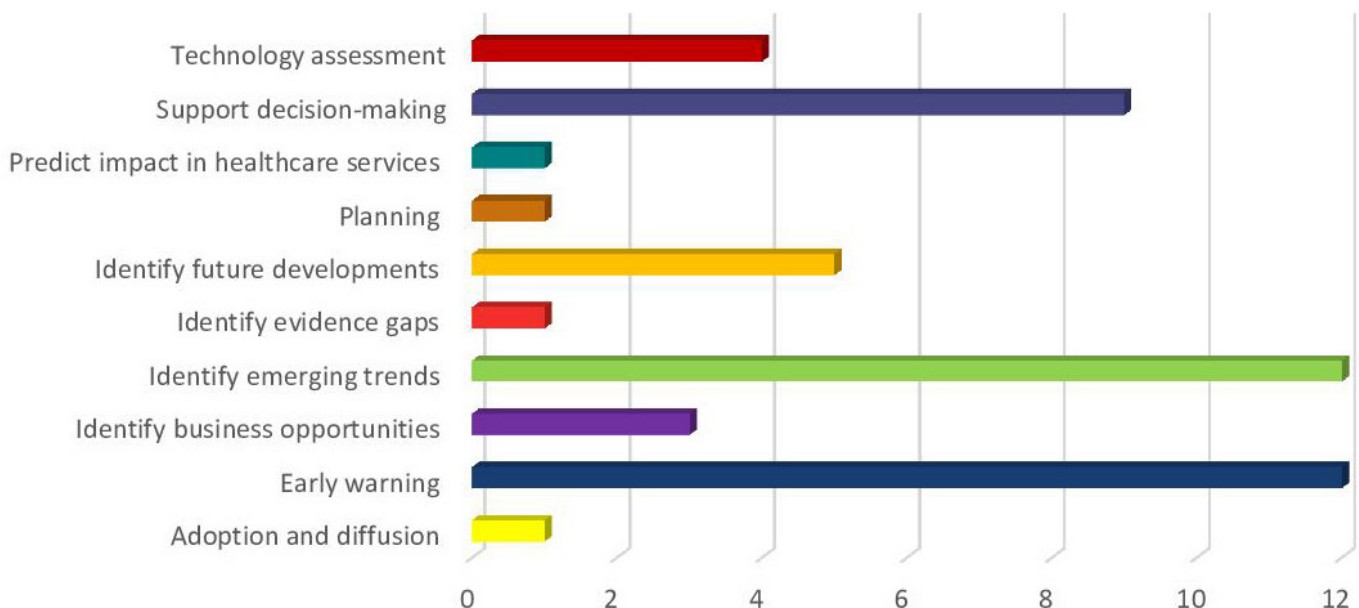

**Figure 1** Aims and objectives for using HS/foresight methods in MedTech innovation identification: early warning;[18–29] identification of emerging trends;[30–41] supporting decision-making;[42–50] identification of future developments;[51–55] supporting technology assessment processes;[56–59] identification of business opportunities;[60–62] predict impact on healthcare services;[63] planning and evaluation;[64] prediction of adoption and diffusion;[65] and identification of evidence gaps.[66] HS, horizon scanning; MedTech, medical technologies.

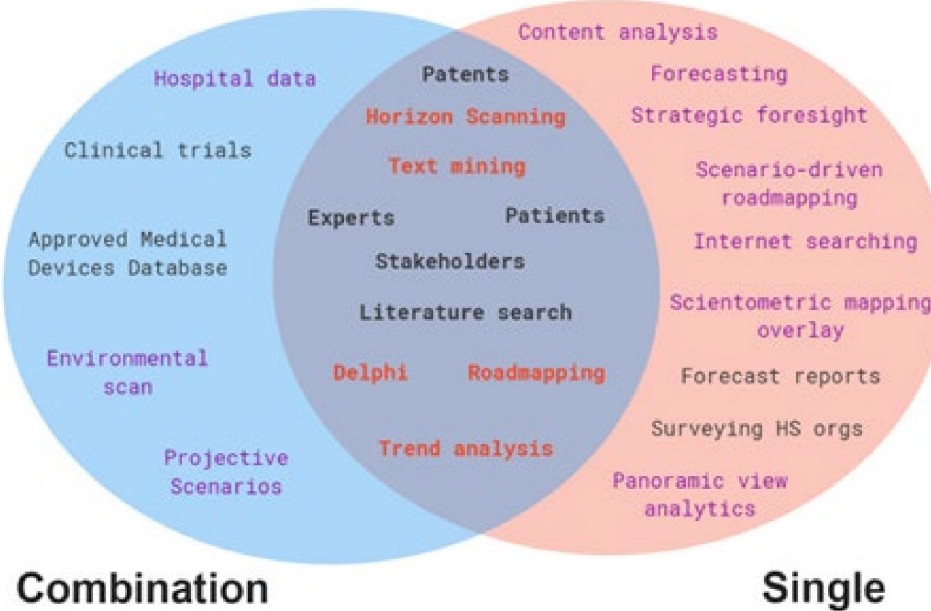

**Figure 2** Methods and sources overlap. Black denotes sources; red denotes methods used by single or combination studies; purple denotes methods only appearing once in any study (combination or single).

Delphi studies, trend analyses, patient engagement and text mining. A small number of uniquely used methods in the combination group encompassed clinical trial analysis, searching approved medical devices databases, hospital data analysis, environmental scan and projective scenarios; while others were uniquely employed by the single-method group: content analysis, forecasting, internet searching, forecast report analysis, panoramic view analytics, scenario-driven road mapping, scientometric mapping approach to overlay mapping, strategic foresight or surveying other HS organisations.

Figure 2 represents a Venn diagram for the overlap of methods as reported; the black writing denotes sources and red and purple denote methods. Methods and sources were often not differentiated. Across both groups, the most frequent and clearly reported method was HS (n=15), followed by surveying or consulting with experts (n=14), literature searching (n=13), patent searching/analysis (n=10), stakeholder engagement (n=4), text/data mining (n=4), trend analysis (n=3) and roadmaps (n=3). Cited less often were report content analyses (n=2),[25 42] Delphi (n=2),[33 57] clinical trial searches (n=2),[47 63] web scraping or mining (n=2),[38 47] and social media searching (n=2).[24 53]

Methods mentioned only once included: environmental scanning,[56] overlay mapping,[34] panoramic view analytics,[66] strategic foresight,[27] surveying companies[23] and surveying other HS organisations.[22]

Forecasting was mentioned by four different reports.[25 32 40 57] One used a combination of expert group opinion forecasting approach, that is, the Delphi technique with a market-oriented capacity analysis by using projective scenarios.[57] The three remaining cite forecasting as a single method using multiple sources including literature and patent searches.[25 32 40]

Less clearly reported methods, often embedded in the reporting of the sources used, were engagement with stakeholders (n=7)[23 26 46 49 56 58 67] and surveying or consulting with patients (n=3).[52 62 68]

Eleven reports published between 2012 and 2020 reported the use of automated methods.[20 24 27 38 40 43 45 47 61 64 66] The most frequently cited was text mining with seven reports describing its use; detailed descriptions of the software packages used were not provided in all instances. Web scraping or URL crawling was described by two reports separately, machine learning (including unsupervised mode) was cited by two reports, data mining was mentioned by two reports and natural language processing by one.

The great majority of the included reports (88%) described the sources of data/intelligence used (figure 3).

Thirteen reports described the use of published literature sources.[28 34 40 47 49–51 54 60 61 63–65] Of these, 3 did not specify whether they also searched unpublished literature,[28 54 60] and a further 12 used published and unpublished sources of literature.[29 31 36 37 39 41 44 46 48 52 53 56]

Literature searching was used in different ways to support the identification of signals. While some reports used it as the basis of the signal identification,[39 41 47 49 52 56 60 65] others used it as a complement to enhance the strength of signals already identified by other means,[36 46 48 50 63] some reports used it in combination with other sources to study relationships[34 40 51 64 69] or expand the signal identification process.[28 29 31 37 44 53 54]

All studies that reported literature searching as a method also used it as a source; however, some studies

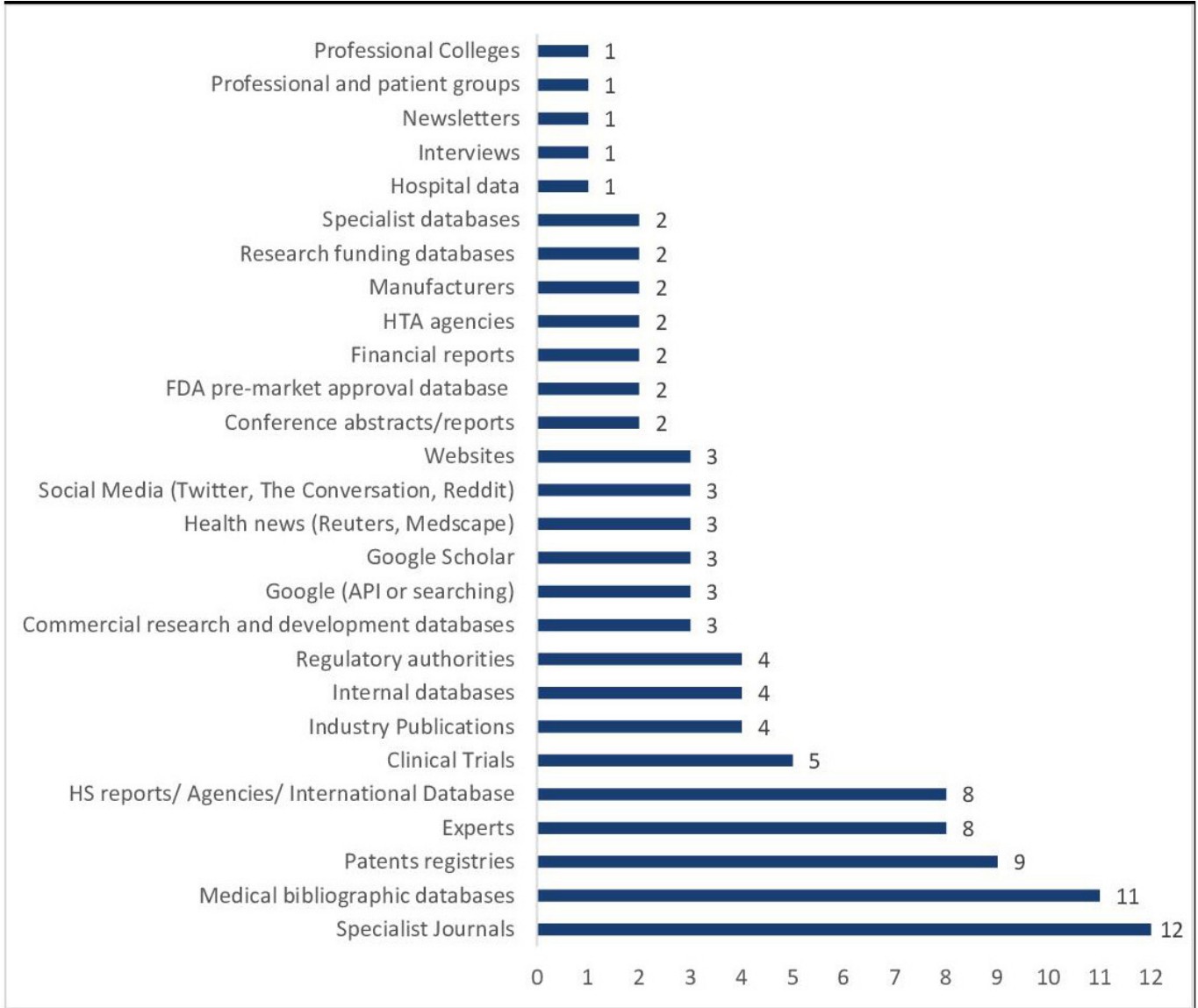

**Figure 3** Type of sources used as described in the included reports. FDA, Food and Drug Administration; HS, horizon scanning; HTA, Health Technology Assessment.

cited literature searching as a source but did not describe it as a method.[28 29 31 34 36 37 39 40 44 56 64]

The analysis of methods reported in the literature, in conjunction with the sources used, identified seven reports[28 29 31 36 37 39 44] that used HS as a single method while searching multiple sources. Three of those reports[29 31 37] additionally practised some level of expert, stakeholder or patient engagement.

From the group of nine papers that reported the use of HS in combination with other methods, seven included consultation with experts, stakeholders or patients.[41 48 55 56 58 62 68] Since half of those studies that reported using HS as a method also reported the use of experts or stakeholders, it cannot be extrapolated that HS has always involved consulting with experts unless explicitly specified.

### Types of technologies included
The most frequently searched for technology were medical devices (n=28), followed by 16 diagnostics, 9

digital health interventions and 3 procedures. Four reports included medical devices, diagnostics and digital health interventions.[44 52 59]

Fifteen reports recorded the inclusion of all technologies but did not provide details and have been analysed as 'all inclusive'.[21 22 25 31 33–35 38 45 47 48 53 57 62 68]

Two reports did not specify any technology within the MedTech scope.[27 29] Online supplemental table 1 presents an overview of the included reports in relation to the technologies in scope and the main purpose of the study.

This review did not identify a relationship between the technologies in scope and the method used for their identification.

### Time horizons
Time horizons were not always described accurately nor was a rationale for the use of a given time limit provided. Some reports mentioned the time horizon as described in the reporting of the searches and often provided both

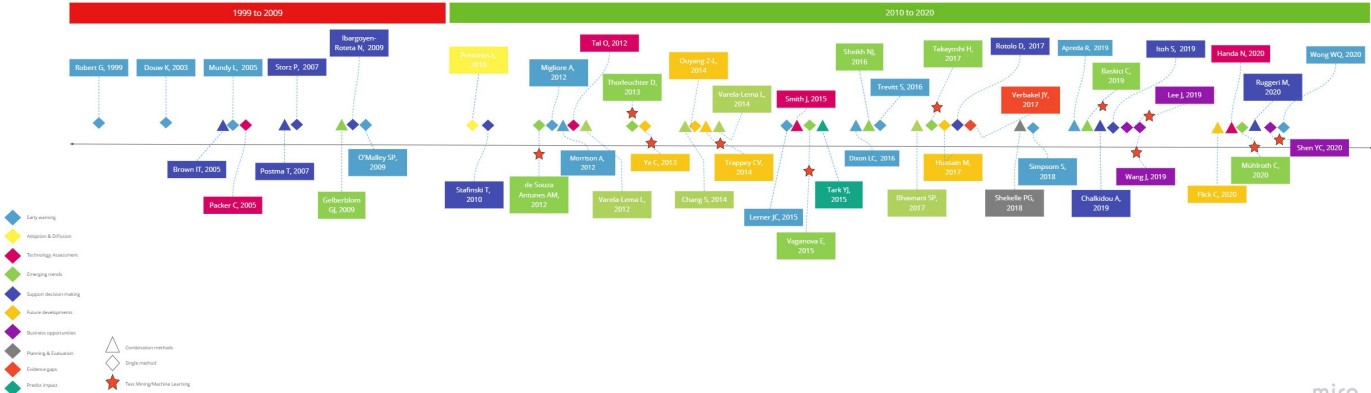

**Figure 4** Development of methods over a publication timeline.

an upper and a lower time limit for technology identification. The time frames were variable and depended on the type of methods used and aim of the study. The longest time frame analysed was 29 years and the shortest 18 months. Longer time frames have been associated with patent mining or searching. Time horizons were used retrospectively or prospectively depending on the time endpoint and aim of the study. For example, retrospective searching was used to identify technologies in scope for established early warning systems[52] or to identify opportunities for future technology development by retrospective searching of patents and analysing results.[51] Prospective methods mostly involved consultation with experts such as clinicians, industry representatives and decision-makers for the prioritisation, prediction or validation of signals.[28 35 53]

As shown in figure 4, 10 studies published between 1999 and 2009 used HS, surveys, a Delphi study and internet searches or a combination of methods mainly consisting of literature searches, expert engagement via interviews and stakeholder surveys. A decade later, the range of methods had exponentially increased, more authors use a combination of methods and these are applied to different decision-making scenarios. Automated and machine learning approaches appear as early as 2012 and underpin almost one-third of the methodologies reported.

## DISCUSSION

This review set out to study how foresight methodologies are used to identify innovative MedTech to support healthcare decision-making. To our knowledge, this review is the first to study the range of foresight methods alongside HS in the context of healthcare decision-making. This review also presents up-to-date information on the sources used for MedTech identification and intelligence analysis in this field. Despite the common understanding that MedTech reports are not usually disseminated by published literature channels, published literature remains as one of the main sources used for

MedTech identification which undermines the ability to discover innovation in a timely fashion.[13]

We discovered some interesting methods that laid outside the limits of the established HS techniques more traditionally defined and used by organisations operating in the health technology space. Methods such as a scientometric mapping approach using overlay mapping of three geographical, social and cognitive spaces[34] provide a visual approach to synthesise complex information and help strategic decision-making. Panoramic view analytics[66] also exploit the benefits of visualisation to explore multidimensional technological information. Scenario-driven roadmapping,[67] a hybrid method developed from traditional roadmapping and scenarios for technology foresight, overcomes some of the limitations of using these methods independently and provides a viable novel method particularly useful in forecasting technology adoption.

The time frames used by the included studies revealed great variability and lacked justification. We recommend the justification of the time frame used in consideration with the life cycles of the technology subject of study (eg, MedTech products 18–24 months). Automation techniques generally used for the identification of signals had been reported within the last decade. There is evidence of their use in patents often in combination with other sources such as clinical trial data and social media platforms such as Twitter or the internet in general. There is potential for application of these techniques to other bodies of data such as news, conference proceedings or multimedia social platforms such as YouTube, but they remain still to be explored for the identification of MedTech signals.

This review has unveiled the broad and vaguely reported use of the term 'horizon scanning' often used as an umbrella to describe the range of methodologies used for the identification of signals, which frequently involve searching in multiple sources and potentially consulting with experts, stakeholders or patients and the public. This inconsistency has been reported previously by Hines *et al*[1] and most recently by Charlton *et al*.[70] Given

the important role that HS plays in health and care policy and decision-making as outlined in the introduction to this study, there is an urgent need for consistency and robust methods standardisation and reporting. We think that HS methods terminology and robust reporting standards will contribute to increase the precision in the identification of published HS literature. An internationally agreed methods guide could provide the groundwork for better standardisation and retrieval of HS reports. The International Horizon Scanning Initiative[71] and the International HealthTechScan[72] are two networks operating in this space that have the potential to contribute towards greater methods standardisation, and the IO has a seat in those two networks to influence and share in enhanced methods initiatives.

Finally, experts, broadly encompassing clinicians, patients and the public, were often used as a method to identify, prioritise or predict technology innovation, as well as a source to compare signals identified by other means. Expert engagement methods were not always specified nor were reported the type of experts (patients, public and professionals) or their professional backgrounds (clinicians, entrepreneurs, regulators, consultancy companies, industry). There is evidence that suggests expert engagement alone can bias predictions;[46] therefore, it would be advisable to practise transparency in the reporting of expert selection and engagement methods.

The use of a systematic literature search and reporting methods to identify published and unpublished literature allowed for a robust and reproducible search in a broad range of databases beyond the traditional medical databases. Further, we believe that this is the first mapping review that aims to discover foresight and HS methods for MedTech innovations to support decision-making. The analysis of objectives, methods and sources together may contribute to answer some of the epistemological questions around which foresight method to use to detect future MedTech development relevant to healthcare decision-makers.[9 73] This review has some limitations. First, the search was geared towards identifying several 'known' methods and that may have limited the possibility of identifying new or unknown methods. However, as already demonstrated, new methods were identified possibly due to the broad inclusion criteria. Second, screening was undertaken mostly by one author with the support of an automated screening software and a second checker. Although some random sampling for screening was undertaken, some potentially valid records could have been missed due to poor indexing or lack of abstract. Concept visualisation of those records could have been used to identify further potentially relevant records; however, due to time and resource constraints, this option was not explored. Lastly, the last search was undertaken in 2021 and no further searches have been run since then. Although it is possible that new methods have been published since, they are unlikely to change the conclusions of this review.

> **Box 1  Recommendations for future methods standardisation**
>
> ⇒ Uniform use of terminology for the description of methods.
> ⇒ Robust reporting of sources searched and strategies used.
> ⇒ Rationale for the use of time limits.
> ⇒ Practise transparency when reporting the background of experts involved.
> ⇒ Report functionality of automated methods when used.

## CONCLUSION AND RECOMMENDATIONS

In the healthcare field, there is global interest in understanding the MedTech innovation pipeline, with a view on system preparedness and acceleration of technology in areas of unmet need. Exploiting HS and foresight methods will support advanced signalling and system preparedness to harness the opportunities posed by innovative MedTech, but also allow us time to address some of the future challenges ahead of those technologies being placed on the market. This review has revealed the use of a range of different methods and tools, used singly or in combination; manually applied or supported with artificial intelligence (text mining and machine learning). These methods have not been applied in a transparent or consistent manner making an assessment of their validity and value challenging. However, they may offer potential solutions to the current methods limitations, and further research in this space should be encouraged. There is also a need to understand the value of alternative sources of unpublished literature such as preprint repositories, conference proceedings or specialist technology-focused sources to mitigate the publication bias already acknowledged. The IO has successfully used preprint repositories for the identification of novel treatments during the recent COVID-19 pandemic[74] and will continue to build on this work. In many areas of research, expert engagement is commonly used. Following an equitable and transparent approach to recruitment and involvement, similar to schemes for patient and public involvement and engagement such as the UK Standards for Public Involvement in Research,[75] should be standard in the practice of foresight. Most importantly, this review clearly highlighted a scientific hindrance due to the lack of methods standardisation and reporting. Based on the findings from this study, some recommendations to improve methods reporting have been suggested in Box 1.

Studying impact of HS intelligence in the decision-making process was out of scope of this review, nevertheless, is an important future research need. This review has provided the foundation work for the next steps in the programme of the National Institute for Health and Care Research Innovation Observatory[76] work which will focus on the development of agreed standards for reporting methods and the development of HS methods guides.

**Contributors** SGG-M was responsible for the conceptualisation of this project, methodology, writing of the original draft, visualisation of results, the overall project administration and is the guarantor of this work. FRB, AOO and LW were involved in the revising and editing of this review and the supervision of this work. CR was involved in the validation of the search strategy, record sifting and writing of original draft. DC was responsible for writing and editing of this review, supervision of this work and funding acquisition.

**Funding** This study/project is funded by the National Institute for Health and Care Research (NIHR) (NIHRIO/project reference HSRIC-2016-10009).

**Disclaimer** The views expressed are those of the author(s) and not necessarily those of the NIHR or the Department of Health and Social Care.

**Competing interests** None declared.

**Patient and public involvement** Patients and/or the public were not involved in the design, or conduct, or reporting, or dissemination plans of this research.

**Patient consent for publication** Not required.

**Ethics approval** This research did not require institutional review board approval as the data were publicly available and collected from existing online databases. This research did not involve any human subjects.

**Provenance and peer review** Not commissioned; externally peer reviewed.

**Data availability statement** All data relevant to the study are included in the article or uploaded as supplemental information.

**ORCID iDs**
Sonia Garcia Gonzalez-Moral http://orcid.org/0000-0003-0431-4771
Luke Wainwright http://orcid.org/0000-0003-3742-6411

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
