## [Reviewer comments · BMJ Open]

ARTICLE DETAILS

TITLE (PROVISIONAL)	Looking at the fringes of MedTech innovation: a mapping review of horizon scanning and foresight methods
AUTHORS	Garcia Gonzalez-Moral, Sonia; Beyer, Fiona; Oyewole, Anne; Richmond, Catherine; Wainwright, Luke; Craig, Dawn

VERSION 1 – REVIEW

REVIEWER	Feiring, Eli University of Oslo
REVIEW RETURNED	08-May-2023

GENERAL COMMENTS	Thank you for this paper. It is interesting, clearly written and is a fine contribution to an understudied subject. I have some comments. P3, line 39 Strengths and limitations: The systematic lack of reporting methods is a limitation in the studies/reports studied – but please describe how this is a limitation of the present review. P4, line 51. Should define/describe horizon scanning as a method – and moreover, explain more explicitly the differences/similarities between HS and foresight methods. P4, line 52. Should perhaps also point out the different types of HS in healthcare settings, i.e. pharmaceuticals, products, devices, care pathways etc. See fex. A Kjelsnes A, Feiring E. Models of integrated care for older people with frailty: a horizon scanning review. BMJ Open 2022;12:e060142. doi:10.1136/bmjopen-2021-060142 P4, line 65: Consider to move the paragraph line 65-82, to line 93. P5, line 86: You claim that a well-conducted HS is, yet this claim is without a reference and it is used in the conclusion P3, line 27 – so, is this a result of the present study or not? P5, line 93: Lacks reference P6, line 122: Was no language limits used? P7, line 142: How did you choose the headings of the Result part? P8, ff The results are presented in both text and figures – is this necessary? Consider how to present some results in the text and the rest in figures. P 31, line 259 ff: What are the implications of the findings? P31, line 269: You claim that HS plays an important role in decision-making yet this claim is not substantiated.
---

	P32, line 275. The Innovation Ob. is first “defined” at the next page. P33, line 305. I am not sure I do agree that this review has revealed that there is a need for, please substantiate these claims further.
--	--

REVIEWER	Sasaki, Hajime The University of Tokyo
REVIEW RETURNED	18-May-2023

GENERAL COMMENTS	This study is a systematic review of horizon scanning for medical technology. As stated in the statement of purpose, horizon scanning has been conducted in many fields in recent years, but in the field of medical technology, it has not been well organized and the overall picture is still unclear. This study is significant as an attempt to conduct a comprehensive survey of these issues. Although I have no objection to the basic significance or the structure of the study, I would like to propose a minor revision to some parts of it. As for the issue of this study, it is stated that no study has examined the effects of applying many HS methods to the MedTech innovation pipeline, but it seems that this issue has not been adequately answered. For example, this issue awareness would be expected to be discussed as to how and when these methods would be effective when used in the regulatory decision-making flow in ICMRA. This research question is a problem statement that is expected to be structured to achieve the essential objectives of horizon scanning, such as what discussions are needed for regulatory purposes from the early stages of development. If possible, we would like to see not only an introduction of the methods, but also a detailed description relating the technology from its manifestation to the regulatory pipeline. In the discussion, there is an impression that the report is limited to typification of methods, and some of the descriptions tend to be superficial. To avoid giving the impression that the word "Medtech" can be replaced with other words, we suggest that the significance of each method be actively presented from the viewpoint of the unique technological or policy aspects of medical technology. While the basic significance is high, the study would be enhanced by including a discussion of practical perspectives that would bring out the best in this study, rather than just introducing the methods.
---

VERSION 1 – AUTHOR RESPONSE

Reviewer 1 (Dr Eli Feiring)	R1, C1: P3, line 39 Strengths and limitations: The systematic lack of reporting methods is a limitation in the studies/reports studied – but please describe how this is a limitation of the present review.	R1, C1: We thank you for taking the time to review our paper. We have addressed this comment by removing this sentence from this bullet point
---	--

	R1, C2: P4, line 51. Should define/describe horizon scanning as a method – and moreover, explain more explicitly the differences/similarities between HS and foresight methods. R1, C3: P4, line 52. Should perhaps also point out the different types of HS in healthcare settings, i.e. pharmaceuticals, products, devices, care pathways etc. See fex. A Kjelsnes A, Feiring E. Models of integrated care for older people with frailty: a horizon scanning review. BMJ Open 2022;12:e060142. doi:10.1136/bmjopen-2021-060142 R1, C4: P4, line 65: Consider to move the paragraph line 65-82, to line 93. R1, C5: P5, line 86: You claim that a well-conducted HS is, yet this claim is without a reference and it is used in the conclusion P3, line 27 – so, is this a result of the present study or not? R1, C6: P5, line 93: Lacks reference R1, C7: P6, line 122: Was no language limits used?	following one of the Editor's comments. R1, C2: Thank you for your suggestion. We have added the definition of horizon scanning in the context of healthcare decision-making by Hines et al. as we have included and referenced their work throughout this review (lines 58 – 59). The differences and similarities between HS and foresight methods are covered in lines 76-80 and then in 86-90. We provide an overview of the similarities and differences of the two families of methods that we include in this review (horizon scanning and foresight methods). R1, C3: We thank you for your reflection on the different types of interventions that HS can help identify. This paper is focussed on the use of HS and foresight methods in MedTech technologies, following this suggestion we have added lines 60 to 63 to address this comment and cited the suggested published article.
--	--	---

	R1, C8: P7, line 142: How did you choose the headings of the Result part? R1, C9: P8, ff The results are presented in both text and figures – is this necessary? Consider how to present some results in the text and the rest in figures. R1, C10: P 31, line 259 ff: What are the implications of the findings? R1, C11: P31, line 269: You claim that HS plays an important role in decision-making yet this claim is not substantiated.	R1, C4: Thank you for this suggestion, we have now moved the paragraph as suggested. R1, C5: Thank you for this observation. We have answered this point by the amended text included in lines 82-86. We have also removed the sentence from the abstract conclusion and replaced it with one that encapsulates this review's findings. R1, C6: Thank you for flagging, this knowledge is substantiated by years of HS practice. We have made the following amendment...to the authors' knowledge there is no evidence to support their use (lines 89-90). R1, C7: The Inclusion Criteria section of this paper establishes that we included papers that were published in English. However, for the search strategy no language limits
--	---	---

	R1, C12: P32, line 275. The Innovation Ob. is first “defined” at the next page. R1, C13: P33, line 305. I am not sure I do agree that this review has revealed that there is a need for, please substantiate these claims further.	were imposed to the search, this means that amongst search results there might have been papers published in languages other than English. This is a common approach that could allow us to understand the size of the non-English literature in this field. Non-English papers were excluded at screening stage in line with the inclusion criteria. The PRISMA flowchart reports these (Appendix C, fig C.1.). R1, C8: The Data Extraction section of this paper outlines the data that was systematically extracted for each one of the included papers. The results section presents an overview of the results in line with the data groups extracted and outlined in the Data extraction section. R1, C9: Thank you for this observation, we have amended Figure 1 to include the references of the included papers by category and removed the redundant text. R1, C10: We have addressed this comment although it is unclear where does this comment end. We have applied further clarification to P31 for timeframes and automation techniques as we understand that P32 onwards the implication are very clear.
--	---	---

		Timeframes: please justify the horizon timeframe or study this in line with the development lifecycle of the technology subject of study, e.g. for MedTech 18 to 24 months ahead of placement in the market. Automation techniques: evidence of use in patents, clinical trial, and social media platforms. Potential for application of these techniques to other bodies of data such as news, conference proceedings or multimedia social platforms such as YouTube remain still to be explored for the identification of signals. R1, C11: The important role that HS plays in decision-making is extensively covered in the introduction section to this paper where paragraph 1 outlines the most recent high-level policies that are putting HS at the heart of much of the decision-making in the UK. We hope that a small reminder included in lines 286-287 would address this comment. R1, C12: Thank you, we have added more information on who the IO is in lines 80 to 83.
--	--	---

		R1, C13: Please see new wording in lines 313-317
Reviewer 2 (Dr Hajime Sasaki)	R2, C1: As for the issue of this study, it is stated that no study has examined the effects of applying many HS methods to the MedTech innovation pipeline, but it seems that this issue has not been adequately answered. R2, C2: For example, this issue awareness would be expected to be discussed as to how and when these methods would be effective when used in the regulatory decision-making flow in ICMRA.	R2, C1: Thank you for your contribution and taking time in reviewing our study. We would like to clarify that to our knowledge, no one has yet provided evidence of which foresight method (including horizon scanning) should be used on each case (lines 89-90). R2,C2: We agree with the reviewer regarding the value of foresight for the regulatory authorities, however this paper has not exclusively framed the value for this decision-maker alone. The aim of this review is to identify as many methods and scenarios for application as possible, to be able to build a methodological framework that would hopefully guide decisions such as the one Dr Sasaki provides as an example 'how and when these methods would be effective when used in the

	R2, C3: This research question is a problem statement that is expected to be structured to achieve the essential objectives of horizon scanning, such as what discussions are needed for regulatory purposes from the early stages of development. If possible, we would like to see not only an introduction of the methods, but also a detailed description relating the technology from its manifestation to the regulatory pipeline. In the discussion, there is an impression that the report is limited to typification of methods, and some of the descriptions tend to be superficial. R2, C4: To avoid giving the impression that the word "Medtech" can be replaced with other words, R2, C5: we suggest that the significance of each method be actively presented from the viewpoint of the unique technological or policy aspects of medical technology. R2,C6: While the basic significance is high, the study would be enhanced by including a discussion of practical perspectives that would bring out the best in this study, rather than just introducing the methods.	regulatory decision-making flow' for a given stakeholder such as the International Coalition of Medicines Regulatory Authorities (ICMRA). It is encouraging to see that Dr Sasaki understands the goal of this programme of work of which this review is just the first step into building a methodological framework. Once this framework is put into practice, it will provide methodological guidance on which is the best fit method to use on innovative MedTech foresight throughout each distinct technology development pipeline. As we state in lines 295-298 'The analysis of objectives, methods and sources together may contribute to answer some of the epistemological questions around which foresight method to use to detect future MedTech development relevant to healthcare decision-makers.' We encourage and would be grateful for Dr Sasaki's willingness to remain in touch with this team as we proceed into the next stage of this research project mapping the different stakeholders needs and foresight demands to support their decision-making including those involved in regulatory decisions of medical devices. R2, C3: We thank Dr Sasaki for sharing his opinion on this regard. The aim of this study is to identify foresight methods applied to MedTech innovation identification or prediction, including but not limited to HS (lines 110-114). The authors
--	---	--

		have felt necessary to include a high-level typification of these methods to aid the understanding of the less expert reader whilst trying not to repeat definitions that have been provided elsewhere by works referenced in this paper. While a more detailed description of each one of the methods identified would have been of value, this would have been challenging to achieve given the heterogeneity and lack of transparency in methods reporting that has been highlighted as a limitation encountered in this study (lines 273-277). R2, C4: Many thanks for your observation with regards the concept of MedTech. We have included (line 91) the range of non-pharmaceutical medical technologies that is commonly considered under the umbrella term MedTech (medical devices, diagnostics and digital applications and systems). We hope that this disambiguation of the term MedTech will be of this reviewer's satisfaction. R2, C5: Table 1 in Appendix D presents a breakdown of methods and technologies in line with their policy or research context. Although we have not been able to establish a robust relationship between the technology and the methods
--	--	---

		used for their identification (lines 227-228), the information in this table will help the reader understand the flexibility in the use of these methods. R2, C6: We have taken this suggestion into consideration and added some recommendations in the conclusion section of this manuscript (lines 317-324).
--	--	---

VERSION 2 – REVIEW

REVIEWER	Feiring, Eli University of Oslo
REVIEW RETURNED	17-Jul-2023

GENERAL COMMENTS	Thank you for the revised manuscript. I have no further comments.
---

REVIEWER	Sasaki, Hajime The University of Tokyo
REVIEW RETURNED	24-Jul-2023

GENERAL COMMENTS	The reviewers' comments have been appropriately responded to and addressed, and the manuscript is deemed worthy of acceptance.
--